# One-Year Follow-Up of Patients Admitted for Emergency Coronary Angiography after Resuscitated Cardiac Arrest

**DOI:** 10.3390/jcm11133738

**Published:** 2022-06-28

**Authors:** Quentin Delbaere, Myriam Akodad, François Roubille, Benoît Lattuca, Guillaume Cayla, Florence Leclercq

**Affiliations:** 1Department of Cardiology, Arnaud de Villeneuve University Hospital, 34295 Montpellier, France; m-akodad@chu-montpellier.fr (M.A.); f-roubille@chu-montpellier.fr (F.R.); f-leclercq@chu-montpellier.fr (F.L.); 2Department of Cardiology, Caremeau University Hospital, 30900 Nîmes, France; benoit.lattuca@chu-nimes.fr (B.L.); guillaume.cayla@chu-nimes.fr (G.C.)

**Keywords:** sudden cardiac death, coronary angiography, neurological sequelae, quality of life

## Abstract

(1) Background: Despite the improvement of the in-hospital survival rate after aborted sudden cardiac death (SCD), cerebral anoxia may have severe neurologic consequences and may impair long-term outcome and quality of life of surviving patients. The aim of this study was to assess neurological outcomes at one year after resuscitated cardiac arrest; (2) Methods: This prospective, observational, and multicentre study included patients >18 yo admitted in the catheterisation laboratory for coronary angiography after aborted SCD between 1 May 2018 and 31 May 2020. Only patients who were discharged alive from hospital were evaluated. The primary endpoint was survival without neurological sequelae at one-year follow-up defined by a cerebral performance category (CPC) of one or two. Secondary end points included all-cause mortality, New York Heart Association (NYHA) functional class, neurologic evaluation at discharge, three-month and one-year follow-up using the CPC scale, and quality of life at 1 year using the Quality of Life after Brain Injury (QOLIBRI) questionnaire; (3) Results: Among 143 patients admitted for SCD within the study period, 61 (42.7%) were discharged alive from hospital, among whom 55 (90.1%) completed the one-year follow-up. No flow and low flow times were 1.9 ± 2.4 min and 16.5 ± 10.4 min, respectively. For 93.4% of the surviving patients, an initial shockable rhythm (*n* = 57) was observed and acute coronary syndrome was diagnosed in 75.4% of them (*n* = 46). At 1 year, survival rate without neurologic sequelae was 87.2% (*n* = 48). Patients with poor outcome were older (69.3 vs. 57.4 yo; *p* = 0.04) and had lower body mass index (22.4 vs. 26.7; *p* = 0.013) and a lower initial Left Ventricle Ejection Fraction (LVEF) (32.1% vs. 40.3%; *p* = 0.046). During follow-up, neurological status improved in 36.8% of patients presenting sequelae at discharge, and overall quality of life was satisfying for 66.7% of patients according to the QOLIBRI questionnaire; (4) Conclusions: Among patients admitted to the catheterisation laboratory for aborted SCD, mainly related to Acute Coronary Syndrom (ACS), less than a half of them were alive at discharge. However, the one-year survival rate without neurological sequelae was high and overall quality of life was good.

## 1. Introduction

Every year, about 350,000 sudden cardiac deaths (SCDs) are reported in the United States and 40,000–50,000 in France, mainly due to acute coronary syndrome (80%) and ventricular fibrillation (VF) [1,2,3]. Few studies have shown the benefit of immediate coronary angiography (CA) in survivors of out of hospital cardiac arrest (OHCA), especially in the setting of ST-segment elevation myocardial infarction (STEMI) [4,5]. The 2017 European Society of Cardiology Guidelines for the management of patients presenting with STEMI recommended direct admission to the catheterisation laboratory (cathlab) in comatose survivors of OHCA with electrocardiographic criteria for STEMI on the post-resuscitation electrocardiogram (ECG) (Class I, grade B) [6]. In the absence of STEMI criteria, admission to an intensive care unit first is recommended to exclude a non-coronary cause (Class IIa, grade B). Unconscious patients admitted to critical care units after SCD are at high risk for death, and neurologic deficits are common among the survivors [7]. Despite improvement in SCD management, the survival rate remains poor [8]. However, for the past few years, the number of immediate CA in patients with OHCA has increased, and prognosis was much more favourable with an 80% survival rate at discharge in a recent report [9]. During SCD, the brain suffers from temporary blood flow limitation, leading to hypoxic brain injury and cognitive impairment [7]. For survivors, the global anoxia can have severe neurological consequences [10]. While the survival rate seems to be well-known, neurological condition of this group of patients is poorly studied [11]. We therefore aimed to assess long-term neurological prognosis of survivors from SCD initially referred to the cathlab for CA.

## 2. Materials and Methods

This study was a prospective, observational, and multicentre registry. First, we checked all patients >18 yo who were admitted directly in the cathlab for out of hospital SCD between 1 May 2018 and 31 May 2020 in Montpellier and Nîmes University Hospitals. Then, we excluded patients who presented ventricular arrhythmia with immediate return of normal consciousness, patients without a return of spontaneous rhythm, or patients who died during the in-hospital stay. Therefore, only patients discharged alive from hospital were included for follow-up analysis. Nimes and Montpellier are university hospitals with intensive care units including cardiac monitoring, where appropriate invasive and non-invasive testing can be performed. A cardiovascular team, including interventional cardiology, electrophysiology, and cardiac surgery, are available. Nearly 150 cardiac arrests are admitted directly in the cathlab per year in these two centres.

The primary end point was the survival rate without significant neurological sequelae at one-year follow-up. Neurological outcome was assessed using the cerebral performance categories (CPC) of the Glasgow-Pittsburgh Outcome Categories: Category 1 is conscious and normal; Category 2 is conscious with moderate disability; Category 3 is conscious with severe disability; Category 4 is coma or vegetative state; and Category 5 is death [12,13,14]. We defined an absence of significant neurological sequelae as conscious, CPC 1 or 2, at one-year follow-up [15]. Secondary end points included total survival and survival without neurological sequelae (CPC 1 or 2) at 3 months, total survival at 1 year, New York Heart Associtation (NYHA) functional class at 3 months and 1 year, and quality of life at 1 year using the Quality of Life after Brain Injury (QOLIBRI) questionnaire. Quality of life was recorded by phone call or mail for SCD survivors without neurological sequelae at one-year follow-up. The QOLIBRI questionnaire is a novel health-related quality of life (HRQoL) instrument specifically developed for traumatic brain injury providing HRQoL in 6 fields [16,17]. This questionnaire has already been used to evaluate quality of life after an SCD [18]. It consists of 37 items in six scales summarised in Figure 1.

Patient characteristics, cardiopulmonary resuscitation (CPR) data (time, location, actors, and methods), and intra-hospital progress were collected at inclusion. If available (by phone call or through medical reports), an initial evaluation was performed at 3 months, with survival rate and cardiac and neurological evaluation. The cardiac and neurological status were also collected at one-year follow-up either by using DxCare software or by a phone call. A questionnaire was offered to all patients at one-year follow-up and was obtained by phone or by mail. At this moment, full study information was given and consent was obtained. No additional testing or biological samples were specifically required for the study.

Considering that our active patient file includes 150 cardiac arrests per year, of which 30 are discharged alive per year, we anticipated the inclusion of 60 patients discharged alive from hospital over a two-year period. Based on a previous study [13], we hypothesised that 10% of patients discharged alive from hospital would have severe neurological sequelae at one-year follow-up. Patient characteristics are presented using mean and standard deviation (SD) for continuous variables and frequencies and proportions for categorical variables. The chi-square test or Fisher’s exact test was used to compare categorical variables between groups (“good” and “poor” outcome). The Student’s *t*-test or the Wilcoxon–Mann–Whitney test was used to compare continuous variables. All analyses were conducted using R software (R Core Team, version 4.0.5, 2021, Vienna, Austria).

## 3. Results

### 3.1. Study Population

A flow chart is presented in Figure 2. A total of 143 patients were admitted directly to the cathlab for aborted SCD after return of spontaneous circulation (ROSC) between May 2018 and May 2020. In-hospital mortality was 57.3% (*n* = 82), 61 patients were therefore alive at discharge, with a mean age of 59.4 ± 14.1 yo and 77.0% (*n* = 47) being men. Baseline characteristics are presented in Table 1. Characteristics of six patients lost to follow-up were not different from our baseline population.

Regarding the cardiovascular risk factors, hypertension was the most frequently observed (*n* = 25, 40.9%). No-flow duration was 1.9 ± 2.4 min, low-flow duration was 16.5 ± 10.4 min, and an initial shockable rhythm was found in 93.4% of patients (*n* = 57). Twenty-nine patients (47.5%) of the survival patients at discharge had an SCD in the presence of a witness: for 18 patients (29.5%) in front of a doctor, for 8 patients (13.1%) in the emergency department, and for 3 patients (4.9%), SCD occurred during a hospital stay. The electrocardiogram immediately after the return of spontaneous circulation mostly showed a STEMI (*n* = 36, 59.0%). The CA found an artery occlusion in 27 patients (44%) and normal status in 15 patients (25%). At discharge, 29 patients (47%) had a left ventricular ejection fraction (LVEF) ≥ 50%. Rehabilitation was offered to all patients at discharge: 30% (*n* = 18), due to medical issues, were directly transferred from hospital to a general rehabilitation centre for neurological improvement, and the others went home to recover before benefiting from cardiac rehabilitation.

Regarding causes of SCD (Figure 3), 75.4% (*n* = 46) were related to ventricular tachycardia or fibrillation resulting from an acute ischemic syndrome. Of these, 78.2% (*n* = 36) were due to a STEMI and 21.7% (*n* = 10) resulted from a severe artery stenosis. Furthermore, 19.7% (*n* = 12) patients presented with a VT or VF resulting from a non-ischemic cardiopathy, 13.1% (*n* = 8) with dilated cardiomyopathy, and 6.6% (*n* = 4) with hypertrophic cardiomyopathy). The last three patients (4.9%) had an initial non-shockable rhythm from an extra-cardiac cause (one stroke, one pulmonary embolism, and one unknown aetiology).

### 3.2. Primary End Point

At one-year follow-up, clinical information was available for 55 patients (reports, phone call, or mail), representing 90.1% of our cohort. Among them, 48 (87.3%) patients were alive without neurologic sequelae as defined previously. A total of three patients (5.5%) died during follow-up and four (7.3%) had severe neurological sequelae (Category 3 on CPC). These seven patients (12.7%) were therefore classified in the “poor event” group (Figure 4).

In the “good outcome” group, six patients (13%) were classified in CPC 2, meaning mild-to-moderate disabilities.

### 3.3. Secondary End Points

Secondary end points are listed in Table 2.

All patients were alive at three-month follow-up. Among the three patients who died at one-year follow-up, none died from cardiovascular causes (two died from sepsis and one from stroke). A total of 19 (35%) patients were hospitalised in the year following the SCD. At discharge, *n* = 6 (10%) patients were classified in CPC 3, *n* = 13 (21%) were CPC 2, and *n* = 42 (69%) were CPC 1. Among patients with CPC 2 or 3, *n* = 7 (37%) recovered during the follow-up: one from CPC 3 to CPC 2 and six from CPC 2 to CPC 1. Cardiac evolution was also favourable with mean LVEF ≥ 50% at 1 year (49.1% at 3 months versus 51.5% at 1 year, *p* = 0.037)). Only three patients (5%) experienced a recurrence of VT/FV and two (4%) suffered a new ischemic event. The number of patients with NYHA functional class I increased from 28 (51%) at 3 months to 36 (69%) at 1 year. We therefore compared characteristics between the two groups named “good outcome” and “bad outcome” (Table 3).

In the “poor outcome” group, patients were significantly older (69.3 yo versus 57.4 yo, *p* = 0.036), with lower BMI (22.4 kg/m^2^ versus 26.7 kg/m^2^, *p* = 0.013), and had lower initial LVEF (32.1 ± 5.7 vs. 40.3 ± 12, *p* = 0.046). Regarding SCD characteristics, there was no difference between the two groups on no-flow or low-flow times and location of initial cardiopulmonary resuscitation (CPR).

### 3.4. Quality of Life

Among the 48 patients contacted to complete the QOLIBRI questionnaire, 36 (75%) returned the mail or accepted to answer by phone call. Results are represented in Table 4. Scores in “satisfaction items” were quite homogeneous with 23.25 ± 6.34 for cognition, 22.14 ± 5 for self, 23.53 ± 7.2 for daily life activities, and 19.56 ± 3.7 for social relationships. In “bothered items”, scores were also similar with 12 ± 4.6 for emotion and 10.9 ± 4 for physical problems.

If we group the answers according to the scale of the questionnaire (from “not at all” to “very”), *n* = 18, 50% of patients were globally satisfied with their quality of life, and *n* = 20, 56% of patients did not report major health problems (Appendix A).

## 4. Discussion

Our study aimed to evaluate the survival rate and the neurological status of patients discharged alive from hospital after an aborted SCD referred for emergency CA with three main findings: (1) patients discharged alive were relatively young, with few cardiovascular risk factors and with an initial shockable rhythm (97%) mainly related to an ischemic aetiology (75%); (2) the survival rate without neurological sequelae at 1 year was high (87%); (3) younger age, lower BMI, and initial better LVEF were associated with survival with good neurological prognosis. For survivors, neurologic and psychological outcome were the main issues with a moderate impairment of quality of life at one-year follow-up.

Acute ischemic aetiology (with or without coronary occlusion) was identified in 75% of patients, and PCI was performed in most of them. We evaluated the outcome of successfully resuscitated patients who were discharged alive. The survival rate without neurological sequelae at 1 year was 87%, which is very encouraging. We selected patients admitted directly to the cathlab with suspicion of ACS and with hemodynamic success of CPR. According to guidelines, only patients with electrocardiographic criteria in favour of ischemic aetiology on the post-resuscitation electrocardiogram (ECG) may have emergency CA in our centres. As we also excluded patients who died during in-hospital stay (57% of patients), our population was highly selective and was relatively young, with normal BMI and a few cardiovascular risk factors. Moreover, time of no and low flow was relatively short, with the presence of a witness in near half of cases, explaining a sub-normal pH on admission. As we expected, almost all of our patients initially presented a shockable rhythm at the time of the SCD. The cardiac aetiology (ischemic and non-ischemic cardiopathy) of the SCD was predominant in our population and an initial shockable rhythm, observed in 93% of our study population, is a well-known strong factor of good prognosis after a cardiac arrest [19,20,21]. Therefore, these results are not generalisable to all patients, especially those with an extra-cardiac cause of SCD.

At 1 year, 87% of our patients were alive without neurological sequelae, which is consistent with previous studies [22,23,24]. In the COACT trial including the population without ST-segment elevation, 64.5% of patients in the immediate CA group were alive at 90 days [25]. An Israelian study found an 85% survival rate at 1 year among patients discharged alive from the intensive cardiology unit [26]. However, beyond mortality, possible anoxic brain injury, mental trauma from surviving a near-death experience, or a new or ongoing cardiac condition can make recovery after SCD difficult. Short-term studies suggest that these complications can lead to an increased physical and psychological burden for both survivors and their relatives [27,28,29]. The neurologic status of patients discharged with severe sequelae did not improve at 1 year (only one patient switched from CPC 3 to CPC 2). On the contrary, half of the patients with “mild-to-moderate” neurologic disabilities (CPC 2) completely recovered at 1 year (CPC 1), also showing progresses in neuro-cognitive care. On the other hand, despite the absence of neurological disorders, quality of life can be somewhat impaired in survivors of SCD. Most patients reported limitations in cognition, self-behaviour, and daily life activities explained by minor physical problems while others could also experience depression or persistent anxiety due to a post-traumatic state [11]. Rehabilitation has therefore been recommended to improve secondary physical and psychological consequences of SCD, but cooperation between cardiological and neurological rehabilitation teams is needed in case of cognitive consequences [30,31]. Despite many improvements in management, more knowledge is needed regarding expectations of OHCA survivors and their relatives. With a similar protocol in a much larger population and with more self-report outcome measurements, the Danish Cardiac Arrest Survivorship (DANCAS survey) will begin. Results will be used to identify the most prevalent problems suffered by OHCA survivors and their families and those at most risk of suffering them [32].

Characteristics of the two groups (good and poor outcome) were similar, except for age, body mass index (BMI), and initial LVEF. Surprisingly, overweight patients had a better neurologic outcome than normal weight patients. Found in previous studies, this phenomenon has been questioned [33,34]. Several explanations of this apparent paradox have been proposed including that higher BMI may allow for the use of higher doses of cardioprotective medications, such as β-adrenergic blockers, particularly in CAD patients, and conversely, lower BMI is traditionally associated with the increase in bleeding events with antithrombotic therapy required after ACS [35,36]. In a recent prospective trial, however, neurological status was assessed in 605 patients resuscitated from SCD. In this cohort, BMI was not associated with good neurologic and survival outcome at discharge [37].

No death during follow-up was due to cardiovascular cause and LVEF was near normal at 1 year, showing the effectiveness of actual management of ischemic heart disease and the beneficial effect of coronary revascularisation. After ACS, cardiac rehabilitation is known to decrease the mortality related to cardiac disease by 20% with the improvement of heart and lung function, socio-psychological status, and the quality of life. In addition, it delays the progression of atherosclerosis and decreases its severity, focusing on education aimed at a healthy lifestyle and improvement of exercise capacity [38,39]. This rehabilitation is safe and well-tolerated in patients with severe comorbidities, such as after aborted SCD [40]. Therefore, for patients without any neurologic sequelae at discharge, apparition of a new cardiac event or recurrence of initial arrhythmia seems not to be the main issue. In a recent study, Kubota et al. showed that the post-discharge mortality of ACS (STEMI or NSTEMI) patients with OHCA was comparable to that of patients without OHCA [41].

The first limitation of our study, even if we collected all patients discharged alive during the inclusion period, remains the small population evaluated. Multivariate analysis was not achievable due to the small number of patients in each subgroup and low incidence of events in the “poor outcome” group. Results of univariate analysis are thus to be taken with caution as we could not adjust with other parameters. Second, the CPC scale is a validated tool in the assessment of neurological status after SCD but it may not be as discriminant as other neurological tools. A CPC score of one or two is commonly regarded as a ‘good outcome’ but it includes subjects with ‘mild-to-moderate’ cognitive impairments, such as dysphasia and permanent memory or mental changes. Furthermore, the CPC seems insensitive to more subtle cognitive impairments. Therefore, the use of a more precise neuropsychological test would be more effective to obtain a precise outcome after SCD [42]. Finally, to assess quality of life, we used the QOLIBRI questionnaire which was initially validated in patients after a traumatic brain injury. Because this work focused on the consequences of neurological sequelae occurring after OHCA, we chose this questionnaire rather than the SF-36 to be more specific. Like others, the accuracy of the statements is uncertain. Moreover, patients’ answers can be modified by many events unrelated to the cardiac arrest. Here, the COVID-19 pandemic may have affected our results with perturbation of daily activities and social relationships, impacting the psychological state, whereas quality of life was good in our population.

## 5. Conclusions

Patients admitted directly to the cathlab after aborted SCD mainly related to ACS had a survival rate of 43% at discharge. For those patients, survival rate without neurological sequelae was excellent at 1 year with 87% of them alive without significant neurological sequelae but with persistent psychological impact in 44% of survivors. These results encourage us to further improve our practices, follow-up methods, and cardiac or multidisciplinary rehabilitation programs.

## Figures and Tables

**Figure 1 jcm-11-03738-f001:**
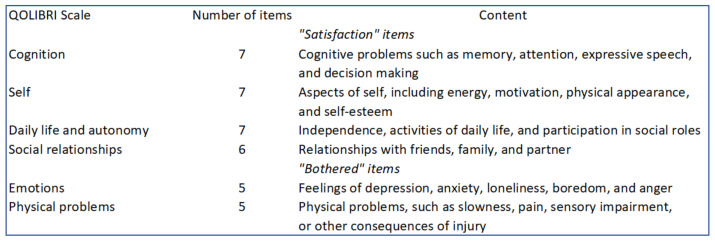
QOLIBRI, Quality of Life after Brain Injury.

**Figure 2 jcm-11-03738-f002:**
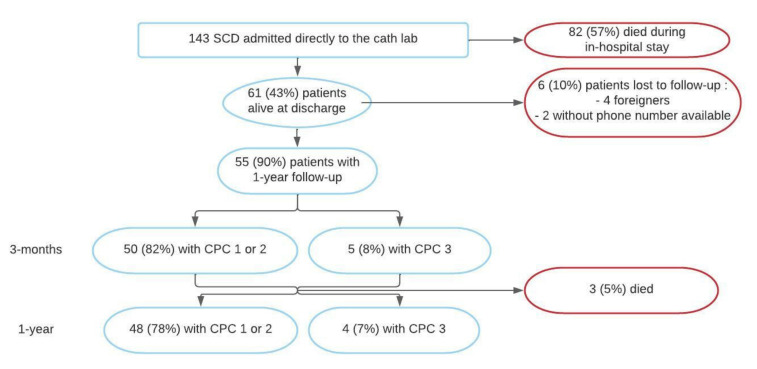
Flow chart. CPC: Cerebral Performance Category; SCD: Sudden Cardiac Death.

**Figure 3 jcm-11-03738-f003:**
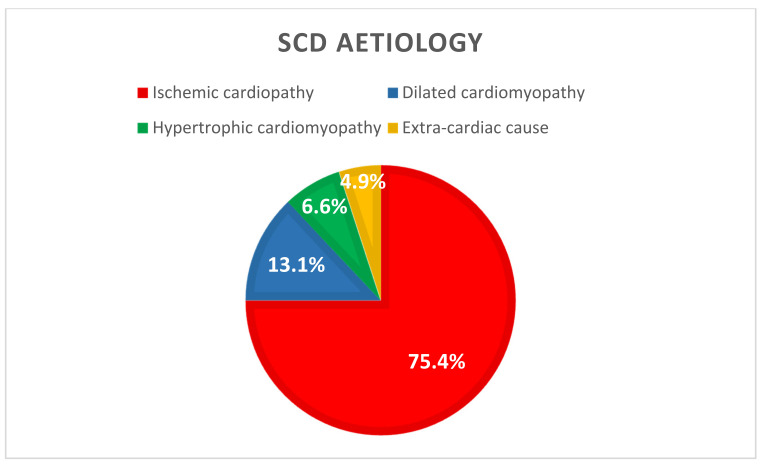
SCD aetiology (*n* = 61). SCD: Sudden Cardiac Death.

**Figure 4 jcm-11-03738-f004:**
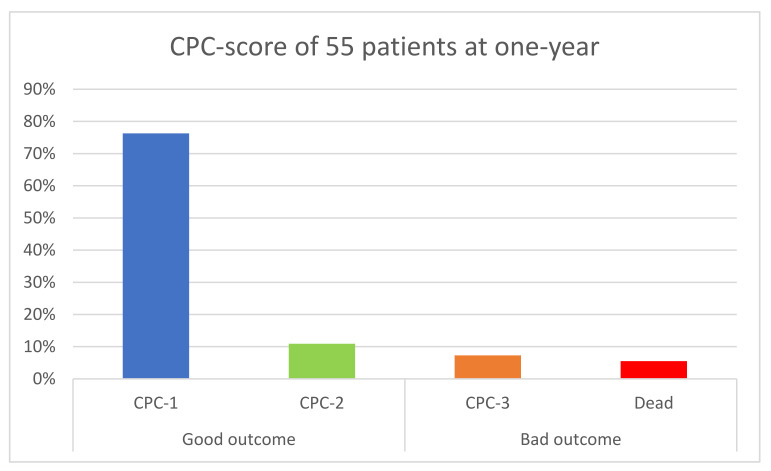
Primary End Point in 55 patients; CPC: Cerebral Performance Category.

**Table 1 jcm-11-03738-t001:** Patients’ study baseline characteristics.

Characteristics *n* (%)	*n* = 61
Age (yo)	59.4 ± 14.1
Male sex	47 (77.0)
Current smoking	19 (31.1)
Hypertension	25 (41.0)
Diabetes	5 (8.2)
Dyslipidemia	11 (18.0)
Body mass index, kg/m^2^	25.9 ± 4.9
History of ischemic cardiopathy	13 (21.3)
Chronic kidney disease	5 (8.2)
**Cardiopulmonary resuscitation:**	
**>Witness**	
Bystander no. (%)	21 (34.4)
Fireman no. (%)	11 (18.0)
Doctor no. (%)	18 (29.5)
Family no. (%)	11 (18.0)
**>Location of SCD**	
Outdoor no. (%)	25 (41.0)
Home no. (%)	25 (41.0)
Emergency department no. (%)	11 (18.0)
**>Timeline**	
No-Flow (min)	1.9 ± 2.4
Low-Flow (min)	16.5 ± 10.4
Total (m)	18.3 ± 11.6
Initial shockable rhythm no. (%)	57 (93.4)
Dose of adrenaline (mg)	1.31 ± 2.08
STEMI on ECG after ROSC no. (%)	36 (59.0)
LVEF after ROSC (%)	35.4 ± 16.1
**>Main coronary lesion**	
Left main coronary no. (%)	1 (1.6)
Left anterior descending no. (%)	25 (41.0)
Left circumflex no. (%)	7 (11.5)
Right coronary no. (%)	13 (21.3)
None	15 (24.6)
**>Biology**	
hs-cTNT (ng/L)	4818 ± 7342
Creatinin peak (µmol/L)	131 ± 109
K^+^ (mmol/L)	4.1 ± 0.66
ASAT (UI/L)	280 ± 260
pH	7.34 ± 0.1
Lactate peak (mmol/L)	± 1.9
**>Evolution**	
Length of ICU stay (d)	7.16 ± 8.32
Length of hospital stay (d)	18.9 ± 11.6
Presence of wall motion abnormalities at entry no. (%)	51 (83.6)
Implantable cardioverter defibrillator no. (%)	19 (31.1)
LVEF > 50% at discharge no. (%)	29 (47.5)
Transfer to rehabilitation centre at discharge no. (%)	18 (29.5)

± Standard Deviation, CA: Cardiac Arrest, ICU: Intensive Cardiac Unit, ROSC: Return of Spontaneous Circulation. LVEF: Left Ventricular Ejection Fraction.

**Table 2 jcm-11-03738-t002:** Secondary end points of 55 patients.

Event	3 Months	1 Year
**Total survival no. (%)**	55 (100)	52 (94.5)
**CPC no. (%)**	*n* = 55	*n* = 52
**1**	43 (78.2)	42 (80.7)
**2**	7 (12.7)	6 (11.5)
**3**	5 (9.1)	4 (7.7)
**Death from cardiac cause no. (%)**	0 (0)	0 (0)
**NYHA**		
**1**	28 (50.9)	36 (69.2)
**2**	27 (49.1)	14 (26.9)
**3**	0 (0)	1 (2)
**4**	0 (0)	1 (2)
**LVEF (%) Mean ± SD**	49 ± 11	51.5 ± 9.2

CPC: Cerebral Performance Category, NYHA: New York Heart Association, LVEF: Left Ventricular Ejection Fraction.

**Table 3 jcm-11-03738-t003:** Univariate analysis.

Characteristic	Good Outcome (*n* = 48)	Poor Outcome (*n* = 7)	*p* Value
Age (y)	57.4 ± 13.6	69.3 ± 9.4	**0.036**
Male sex no. (%)	35 (72.9)	7 (100)	0.18
Body mass index kg/m^2^	26.7 ± 5	22.4 ± 2.2	**0.013**
Current smoking no. (%)	14 (29.2)	3 (42.9)	0.66
Hypertension no. (%)	20 (41.7)	4 (57.1)	0.69
Diabetes no. (%)	5 (10.4)	0 (0)	1
Dyslipidemia no. (%)	10 (20.8)	1 (14.3)	1
CKD no. (%)	4 (8.3)	1 (14.3)	0.51
Low flow (min)	16.4 ± 10.7	17 ± 9.3	0.72
No flow (min)	1.9 ± 2.5	2 ± 2.2	0.68
Total CPR (min)	18.3 ± 11.8	19 ± 11	0.65
LAD culprit no. (%)	18 (37.5)	4 (57.1)	0.78
STEMI no. (%)	28 (58.3)	4 (57.1)	0.5
Occlusion no. (%)	20 (41.7)	3 (42.8)	0.88
Outdoor no. (%)	19 (39.6)	4 (57.1)	0.22
Home no. (%)	22 (45.8)	1 (14.3)	0.22
Witness no. (%)	15 (31.3)	4 (57.1)	0.45
Initial shockable no. (%)	44 (91.7)	7 (100)	1
Length of hospital stay (d)	17.6 ± 12.5	31.7 ± 28.6	0.11
Length of reanimation stay (d)	7.5 ± 8.8	7.4 ± 7.7	0.93
ASAT (UI/L)	282 ± 240	356 ± 430	1
K^+^ (mmol/L)	4 ± 0.6	4.5 ± 0.7	0.16
Lactate peak (mmol/L)	2.8 ± 2	3.5 ± 2	0.23
hs-cTNT (ng/L)	4362 ± 6363	9417 ± 13,903	0.98
Creatinin peak (µmol/L)	140 ± 120	107 ± 29.3	0.88
Initial LVEF (%)	40.3 ± 12	32.1 ± 5.7	**0.046**
LVEF at discharge (%)	48.4 ± 11	41.4 ± 14.1	0.2
LVEF at 3 months (%)	50.1 ± 10.6	42.9 ± 12.5	0.13
LVEF at 1 year (%)	51.9 ± 9.3	46.3 ± 4.7	0.19

CKD: Chronic Kidney Disease (Glomerular Fraction Rate < 50 mL/min/1.73 m^2^), CPR: Cardiopulmonary Resuscitation, ICD: Implantable Cardiac Device, LAD: Left Anterior Descending, STEMI: ST-segment Elevation Myocardial Infarction, AST: Aspartate Transaminase, LVEF: Left Ventricular Ejection Fraction. Bold: *p* < 0.05, statistically significant.

**Table 4 jcm-11-03738-t004:** Mean score for each item of QOLIBRI.

Quality of Life (QOLIBRI)	Mean Score ± SD	Range of the Questionnaire
**Part I: Satisfaction**	88.5 ± 20	[20-140]
Cognition	23.25 ± 6.3	[5-35]
Self	22.14 ± 5	[5-35]
Daily life activities	23.53 ± 7.2	[5-35]
Social relationships	19.56 ± 3.7	[5-25]
**Part II: Problems and complaints**	22.9 ± 8.3	[10-40]
Emotion	12 ± 4.6	[5-20]
Physical problems	10.9 ± 4	[5-20]

## Data Availability

Datas are available in department of cardiology, University of Montpellier, France.

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
