# Peer review of "One-Year Follow-Up of Patients Admitted for Emergency Coronary Angiography after Resuscitated Cardiac Arrest"

_jcm, 2022, doi:10.3390/jcm11133738_

Round 1

Reviewer 1 Report

Thank you for the opportunity to review the paper by Delbaere et al on the one-year follow up of patients discharged after resuscitated sudden cardiac death. Amongst the 61 patients who were discharged alive, 55 completed one-year follow up. Majority had good neurological outcomes at 1-year, though almost half reported psychological issues.

My concerns are listed below:

Abstract

11.   The title and abstract are somewhat misleading in that the study population seemed to be those who survived to hospital admission rather than survived to hospital discharge. For example, in the abstract, the authors stated “Among 143 patients included within study period….” and that gave the impression that was the study population.

22.  The aim of the study is missing from the abstract.

Methods:

11.  Inclusion and exclusion criteria are missing. Did the study include only out-of-hospital SCD, or both out- and in-hospital SCD?

22.   Please include more information about the study setting. Did the study include patients from 2 University Hospitals? If so, how big is these hospitals and what is the usual volume of SCD? Along the same lines, what is the usual post-resuscitation care in these hospitals?

33.  The authors explained what CPC is and the different scales (ie 1-5). CPC 1 and 2, especially CPC 2, should not be considered no neurological sequelae.

44.   What are DxCare and Clinicom? Please provide elaborations on that.

55.   Please elaborate on how clinical evaluation was conducted at 3 months.

66.  There are too many secondary endpoints. I would suggest being selective and focusing only on 3 important ones.

Results:

11.   In Table 1, what does “Actors” mean?

22.  Given there are 3-month evaluation and 1-year evaluation, findings can be better illustrated with the use of a figure.

33.  The drop-out rate is 10% at 1 year. Are these drop-outs different from those who remained in the study? This was not mentioned in the paper.

Discussion:

11.       Key limitations are left out: This appears to be a highly selected group of presumed cardiac etiology (hence the immediate coronary angiography) and findings won’t be generalizable to all SCD patients, response bias should be considered particularly in HRQOL findings.

Reviewer 2 Report

The present study was designed to evaluate aimed to evaluate the survival rate and the neurological status of patients discharged alive from the hospital after an aborted sudden cardiac death and referred for emergency coronary angiography.

In the abstract, it is not clear what the authors intended to say bay "duration of no and low-flow period was 1.9 2.4 and 16.5 10.4 minutes, respectively"

The inclusion criteria should be better defined in the Material and Methods section.

"Figure 4. Primary End Point in 55 patients. CPC: Cerebral Performance Category." should have a description so that the readers to better understand the information in it.
